# Adverse events following immunization with ChAdOx1 nCoV-19 and BBIBP-CorV vaccine: A comparative study among healthcare professionals of Nepal

Sushil Rayamajhi[1,2], Md. Abdur Rafi[2,3], Nishant Tripathi[4], Anjana Singh Dongol[5], Minalma Pandey[1], Shreejana Rayamajhi[6], Subhechchha Bhandari[7], Pranay Shrestha[8], M. Tasdik Hasan[9,10,11], Md. Golam Hossain[12]*

1 Swacon International Hospital, Kathmandu, Nepal, 2 Rajshahi Medical College, Rajshahi, Bangladesh, 3 Pi Research Consultancy Center, Dhaka, Bangladesh, 4 Department of Medicine, University of Kentucky Medical Center, Lexington, Kentucky, United States of America, 5 Kathmandu University School of Medical Science, Dhulikhel, Nepal, 6 Master of Science in Digital Health, Deggendorf Institute of Technology (Technische Hochschule Deggendorf), Deggendorf, Germany, 7 Department of International Community Health, University of Oslo, Oslo, Norway, 8 Department of Information Management, Tribhuvan University, Kirtipur, Nepal, 9 Jeeon Bangladesh Ltd., Dhaka, Bangladesh, 10 Public Health Foundation, Bangladesh (PHF, BD), Dhaka, Bangladesh, 11 Department of Primary Care & Mental Health, University of Liverpool, Liverpool, United Kingdom, 12 Department of Statistics, Health Research Group, University of Rajshahi, Rajshahi, Bangladesh

* hossain95@yahoo.com

**Editor:** Kovy Arteaga-Livias, Hermilio Valdizán National University Academic Professional School of Medicine: Universidad Nacional Hermilio Valdizan Escuela Academico Profesional de Medicina Humana, PERU

## Abstract

### Background

Adverse events following immunization (AEFI) against SARS-CoV-2 are common as reported by clinical trials and contemporary evidence. The objective of the present study was to evaluate the local and systemic adverse events following vaccination with ChAdOx1 nCoV-19 and BBIBP-CorV among the healthcare professionals (HCPs) of Nepal.

### Methods

This cross-sectional study was conducted among 606 vaccinated HCPs of Kathmandu, Nepal. Data was collected from June 15 to 30, 2021 using a self-administered online survey tool. Multiple binary logistic regression models were used to predict the adverse events according to the vaccine types and doses after adjusting for age, sex, comorbidity and previous SARS-CoV-2 infection.

### Results

The mean (SD) age of the participants was 35.6 (13.2) years and 52% of them were female. Almost 59% of participants were vaccinated with two doses and around 54% of total of them took the ChAdOx1 nCoV-19 vaccine. At least one local and systemic adverse event was reported by 54% and 62% of participants after the first dose and 37% and 49% after the second dose of ChAdOx1 nCoV-19 and by 37% and 43% after the first dose and 42% and 36%

**Data Availability Statement:** All relevant data are within the manuscript and its Supporting Information files.

**Funding:** The authors received no specific funding for this work.

**Competing interests:** The authors have declared that no competing interests exist.

**Abbreviations:** AEFI, Adverse events following immunization; HCP, Health care professional; WHO, World Health Organization; cOR, Crude odds ratio; aOR, Adjusted odds ratio; CI, Confidence interval.

after the second dose of BBIBP-CorV vaccine respectively. Injection site pain, swelling and tenderness at the injection site were the most frequently reported local AEFI while, fatigue, headache, fever and myalgia were the most frequently reported systemic AEFI. The logistic model demonstrated that the risk of both local and systemic adverse events was higher among the ChAdOx1 nCoV-19 vaccine recipients compared to the BBIBP-CorV vaccine. Almost 10% of individuals reported a post-vaccination SARS-CoV-2 infection and most of them occurred after taking the first dose of vaccine.

## Conclusions

Recipients of both the ChAdOx1 nCoV-19 and BBIBP-CorV vaccine among the HCPs of Nepal reported only mild and constitutional symptoms including injection site pain and tenderness, headache, fever, fatigue, etc. after vaccination.

## Introduction

The world is currently facing the largest pandemic of the century due to a newly emerged highly transmissible and pathogenic virus of the coronavirus family, the SARS-CoV-2 which causes an acute respiratory disease, named 'coronavirus disease 2019' (COVID-19) [1]. This pandemic contributed to more than 205 million infections by the virus and four million deaths worldwide [2]. Vaccination against SARS-CoV-2 is recognized as a leading strategy to combat the pandemic and several vaccines have been promptly developed and approved by different health authorities for mass vaccination worldwide [3, 4].

In Nepal, almost 721 thousand COVID-19 cases were reported with almost 10000 deaths by June 2021 [2]. The government of Nepal started a public vaccination program in late January 2021 and administered more than 4.5 million doses of ChAdOx1 nCoV-19 and BBIBP-CorV vaccine by June 2021 covering 8% of their total population [5]. The ChAdOx1 nCoV-19 is an adenoviral vector vaccine manufactured by the Oxford-AstraZeneca, UK and BBIBP-CorV is an inactivated SARS-CoV-2 vaccine manufactured by the Sinopharm, China [3]. The large-scale clinical trials reported adequate safety profiles and antibody responses by both of the vaccines [6–9].

The clinical trials, as well as real-world evidence, reported some adverse events following immunization (AEFI) with these vaccines though most of these were mild constitutional symptoms. A population-based study from the UK conducted among more than half a million recipients of the BNT162b2 mRNA vaccine (Pfizer) or ChAdOx1 nCoV-19 recombinant vaccine (Oxford-AstraZeneca) reported that almost 22% of individuals after taking BNT162b2 vaccine, and 34% individuals after taking the ChAdOx1 nCoV-19 vaccine developed at least one systemic symptom, mostly mild headache, fatigue, and fever. Local symptoms like injection site pain, tenderness, and swelling were reported by almost half of the recipients [10]. Severe adverse events like anaphylaxis or thromboembolism were very rare after vaccination against SARS-CoV-2 [11–13].

Despite the facts, fear of side effects remains one of the major contributors to the COVID-19 vaccine hesitancy worldwide [14, 15]. Reports on vaccine side effects from own community could make the people confident about the vaccine. Moreover, there is hardly any data on the real-world side effects of ChAdOx1 nCoV-19 and BBIBP-CorV vaccine among the South-East Asian population. Healthcare professionals, especially physicians, nurses, and medical

technicians can be a suitable group for reporting the side effects of vaccination due to their professional knowledge and trust in the population. Hence, the present study aims to report the local and systemic adverse events experienced following immunization with these two major vaccines deployed among the healthcare professionals (HCPs) of Nepal.

## Methods

### Study design and participants

This cross-sectional study was conducted among the HCPs working within the geographical area of Nepal and who took at least one dose of the ChAdOx1 nCoV-19 or BBIBP-CorV vaccine between February 15 and May 31, 2021, from different vaccination centers of Nepal. The sample size was calculated using the following formula:

$n = \frac{z^2 p(1-p)}{d^2}$, where, p = prevalence of side effects after vaccination, and d = precision of error. Menni et al. reported that 33·7% of the ChAdOx1 nCoV-19 vaccine recipients developed at least one side effect [10]. Using this information the calculated sample size was 343. The convenience sampling method was used to include participants. An online survey tool was used for data collection. Five physicians, three nurses, and three medical technologists were recruited to circulate the survey link among their professional networks through social media (Facebook). The survey link was also posted in different closed groups of these professionals with a description of the aims of the study and a request to fill it up. Finally, a total of 606 HCPs (386 physicians and 238 other staff such as nurses and medical technicians) filled up the form completely.

Data collection procedure: An online survey using Google Forms created with the guidance of previously published articles [10, 16, 17] was used for data collection. It had three parts: (i) the socio-demographic characteristics of the participants, (ii) self-reported side effects after vaccination, and (iii) history of SARS-CoV-2 infection before or after vaccination. If SARS-CoV-2 RNA or antigen was detected on a respiratory specimen collected $\geq$14 days after completing the primary series of the vaccines, it was defined as a vaccine breakthrough infection [18].

### Ethical consideration

The ethical clearance of the research was obtained from the Nepal Health Research Council (NHRC) (Reference no. 351). Written consent was taken through email from the participants who agreed to participate in the study as explained on the first page of the survey form.

### Statistical analysis

All statistical analyses were conducted using the STATA version 16.0. Frequency distribution with percentage was used to report the side effects experienced by the participants. The Chi-square test was used to determine the association between two categorical variables. Finally, multiple binary logistic regression adjusted for age, sex, comorbidity, and previous SARS-CoV-2 infection was used to predict the side effects according to the vaccine types and doses.

## Results

A total of 606 HCPs (386 physicians and 238 supporting staff such as nurses and medical technicians) were considered as the sample in this study. The mean (SD) age of participants was 35.6 (13.2) years and almost 52% of them were female. More than 32% of the participants were infected by SARS-CoV-2 before vaccination. 17.66% of participants had comorbidity (such as type 2 diabetes mellitus, hypertension, chronic respiratory diseases like asthma and COPD,

**Table 1. Socio-demographic characteristics of the participants (n = 606).**

| Characteristics | Total, N(%) | Single dose, Total = 250; N(%) | Two dose, Total = 356; N(%) |
|---|---|---|---|
| Age (years) (mean 35.63, SD 13.17) | | | |
| <55 | 515 (84.98) | 188 (32.30) | 327 (56.19) |
| ≥55 | 91 (15.02) | 62 (68.13) | 29 (31.87) |
| Sex | | | |
| Male | 290 (47.85) | 126 (39.25) | 164 (51.09) |
| Female | 316 (52.15) | 124 (35.23) | 192 (54.55) |
| Profession | | | |
| Physician | 386 (63.70) | 115 (29.11) | 253 (64.05) |
| Supporting stuffs | 238 (39.27) | 135 (48.56) | 103 (37.05) |
| Comorbidity | | | |
| Yes | 107 (17.66) | 62 (55.86) | 45 (40.54) |
| No | 499 (82.34) | 188 (33.45) | 311 (55.34) |
| Previous SARS-CoV-2 infection | | | |
| Yes | 194 (32.01) | 74 (36.10) | 120 (58.54) |
| No | 412 (67.99) | 176 (37.61) | 236 (50.43) |
| Vaccine type | | | |
| ChAdOx1 nCoV-19 | 328 (54.13) | 101 (30.79) | 227 (69.21) |
| BBIBP-CorV | 278 (45.87) | 149 (53.60) | 129 (46.40) |

history of cardiovascular diseases, etc.). It was observed that around 54% of participants took the ChAdOx1 nCoV-19 vaccine and 46% took the BBIBP-CorV vaccine. The type of vaccine was decided by the government authority based on availability. Almost 59% of participants took both doses of vaccine (Table 1).

The overall prevalence of local adverse events was 45% and 44% and the prevalence of systemic adverse events was 54% and 44% after the first and second dose respectively. It was observed that both local and systemic adverse events were more prevalent after both first and second doses of ChAdOx1 nCoV-19 compared to BBIBP-CorV (54% vs 37% and 62% vs 43% respectively after the first dose and 45% vs 42% and 49% vs 36% respectively after the second dose). Besides, these adverse events occurred more frequently after the first dose than the second dose of both vaccines. Pain, swelling and tenderness at the injection site were the most frequently reported local adverse events after both the first and second dose of the vaccines. On the other hand, fatigue, headache, fever and myalgia were the most frequently reported systemic adverse events (Table 2). No serious and life-threatening adverse event like anaphylaxis was reported.

Adverse events experienced by the participants after both vaccines were homogenous in different socio-demographic groups of participants. The logistic regression model showed that both local and systemic adverse events after the first dose of vaccination occurred at a comparatively higher rate in the case of the ChAdOx1 nCoV-19 vaccine (aOR 1.81, 95% CI 1.29–2.54 for local adverse events and aOR 2.32, 95% CI 1.65–3.26 for systemic adverse events) while this trend was observed for only systemic adverse events after the second dose (aOR 1.63, 95% CI 1.02–2.58). On the other hand, no significant difference was observed in the case of local adverse events of the ChAdOx1 nCoV-19 vaccine after the first and second dose, while systemic adverse events were more likely after the first dose of the vaccine (aOR 1.76, 95% CI 1.24–2.49). In the case of the BBIBP-CorV vaccine, both local and systemic adverse events were more likely after the first dose (aOR 2.61, 95% CI 1.57–4.35 for local and aOR 3.55, 95% CI 2.08–6.05 for systemic adverse events) (Fig 1).

**Table 2. Adverse events reported by the participants after vaccination (n = 606).**

| Adverse events | ChAdOx1 nCoV-19 | | | | BBIBP-CorV | | | |
|---|---|---|---|---|---|---|---|---|
| | First dose | | Second dose | | First dose | | Second dose | |
| | n | % | N | % | N | % | n | % |
| Local adverse events | | | | | | | | |
| Any local adverse event | 178 | 54.3 | 102 | 44.9 | 102 | 36.7 | 54 | 41.9 |
| Pain at injection site | 130 | 39.6 | 64 | 28.2 | 76 | 27.3 | 32 | 24.8 |
| Swelling | 64 | 19.5 | 36 | 15.9 | 36 | 12.9 | 14 | 10.9 |
| Tenderness | 46 | 14.0 | 47 | 20.7 | 24 | 8.6 | 22 | 17.1 |
| Itch | 8 | 2.4 | 0 | 0.0 | 6 | 2.2 | 0 | 0.0 |
| Swollen armpit gland | 3 | 0.9 | 0 | 0.0 | 4 | 1.4 | 0 | 0.0 |
| Redness | 22 | 6.7 | 18 | 7.9 | 10 | 3.6 | 14 | 10.9 |
| Warmth | 29 | 8.8 | 22 | 9.7 | 8 | 2.9 | 2 | 1.6 |
| Systemic adverse events | | | | | | | | |
| Any systemic adverse event | 205 | 62.5 | 112 | 49.3 | 120 | 43.2 | 46 | 35.7 |
| Headache | 109 | 33.2 | 66 | 29.1 | 71 | 25.5 | 20 | 15.5 |
| Fatigue | 144 | 43.9 | 80 | 35.2 | 71 | 25.5 | 31 | 24.0 |
| Chills and shivering | 32 | 9.8 | 16 | 7.0 | 8 | 2.9 | 6 | 4.7 |
| Diarrhea | 10 | 3.0 | 4 | 1.8 | 5 | 1.8 | 0 | 0.0 |
| Fever | 108 | 32.9 | 80 | 35.2 | 55 | 19.8 | 23 | 17.8 |
| Arthralgia | 8 | 2.4 | 30 | 13.2 | 10 | 3.6 | 4 | 3.1 |
| Myalgia | 41 | 12.5 | 37 | 16.3 | 6 | 2.2 | 8 | 6.2 |
| Nausea | 22 | 6.7 | 14 | 6.2 | 14 | 5.0 | 2 | 1.6 |
| Rash | 0 | 0.0 | 4 | 1.8 | 0 | 0.0 | 0 | 0.0 |

A total of 61 participants were infected with SARS-CoV-2 after vaccination (infection rate 10%). Among them, 52 cases were reported after the first dose of vaccination (27 after the ChAdOx1 nCoV-19 vaccine and 25 after the BBIBP-CorV vaccine). On the other hand, all the 9 cases were reported after the second dose of the ChAdOx1 nCoV-19 vaccine. The median window period between vaccination and infection after the first dose was 13 days (range 3 to 43 days) and between vaccination and infection after the second dose was 6 days (range 3 to 12 days) (Fig 2). No vaccine breakthrough infection which is defined as infection by SARS-CoV-2 after at least 14 days of completion of two-dose vaccination was reported in the study.

## Discussion

In this study, we investigated the adverse events experienced by the HCPs of Nepal following the administration of the two COVID-19 vaccines that are in use in Nepal. These two vaccines are being used by many South Asian countries so the findings are very relevant and important in this mass vaccination phase. According to our findings, more than half of the participants experienced mild (to moderate) local and systemic adverse events after vaccination.

At least one local and systemic adverse event was reported by almost 54% and 62% of participants after the first dose and 37% and 49% of participants after the second dose of ChAdOx1 nCoV-19 vaccine respectively in our study. A large-scale community-based study from the UK reported that local and systemic side effects occurred in 59% and 34% of participants respectively after the first dose of the same vaccine [10]. Though the prevalence of local adverse events was somewhat similar, that of systemic adverse events was comparatively higher in our study. However, it was lower than the expected rate of 88% as reported by a phase-3 trial of the vaccine [7]. Prevalence of common local and systemic adverse events like injection site pain

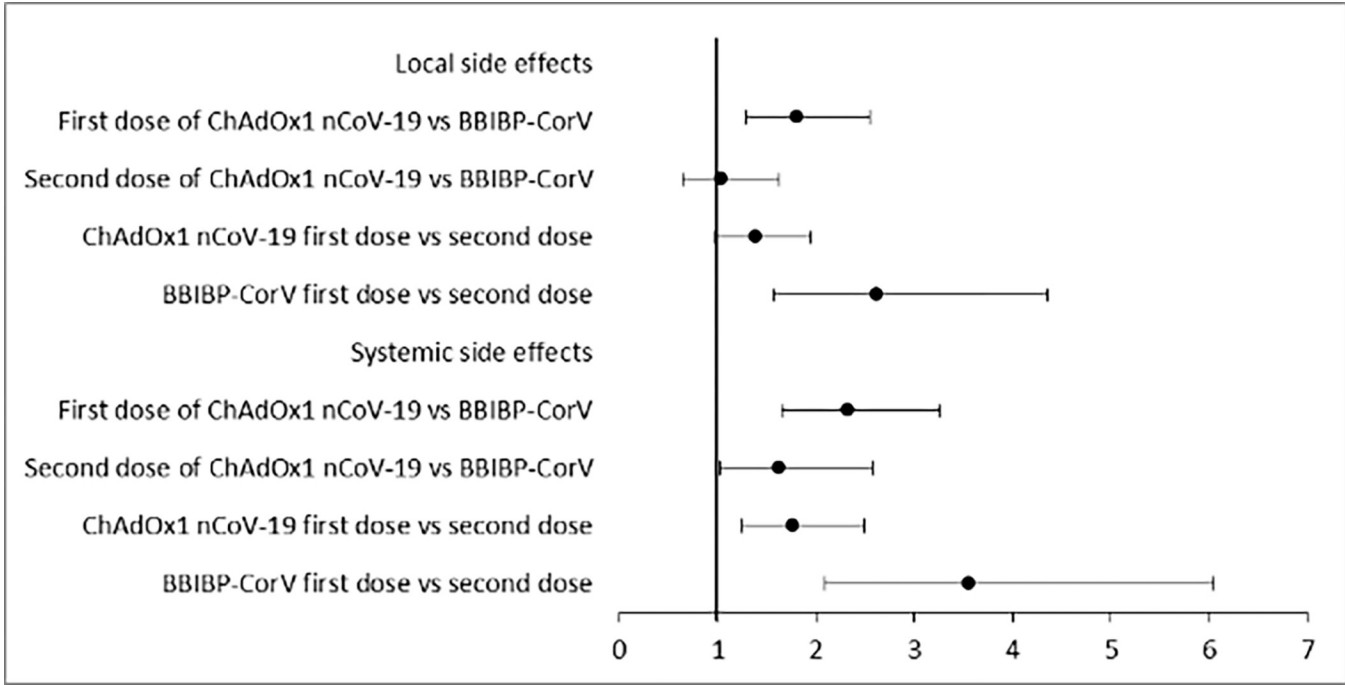

**Fig 1. Risk of adverse events according to vaccine type and doses (OR with 95% CI adjusted for age, sex, comorbidities, and previous SARS-CoV-2 infection).**

and tenderness, headache, fever, fatigue, etc. were reported lower among the British population [10], and comparatively higher in Iraqi and Korean populations [16, 17] than in our study.

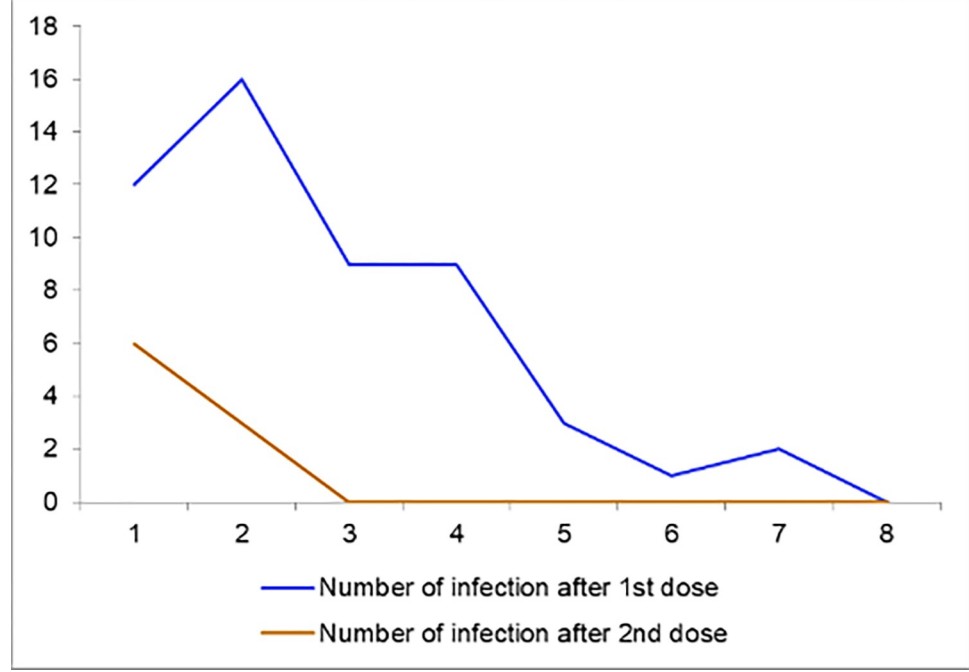

**Fig 2. SARS-CoV-2 infection among the participants up to 8 weeks after vaccination (n = 61).**

In the case of the BBIBP-CorV vaccine, local and systemic adverse events were reported by almost 37% and 43% respectively after the first dose and 42% and 36% respectively after the second dose. These findings were quite similar to the reports of the phase-2 trials of the vaccine, where almost one-third of the vaccine recipients experienced at least one adverse event [9, 19]. Injection site pain (22%) was the most common adverse event in the trial followed by fever (4%) and fatigue (3%) [19]. Though the prevalence of these adverse events was higher among our study participants compared to the trials, real-world evidence from Iraq reported a higher prevalence of these symptoms among the BBIBP-CorV vaccine recipients [16].

Our study revealed that the odds of both local and systemic adverse events were higher among the ChAdOx1 nCoV-19 vaccine recipients compared to the BBIBP-CorV vaccine. Evidence of comparative prevalence of adverse events of these two vaccines is scanty. One study conducted in Iraq supported our findings [16]. We could not establish any significant association between vaccine adverse events and the age or sex of the recipients, though prevalence was slightly higher among females. A higher prevalence of adverse events among females was also reported by some studies, though the role of age remained ambiguous [10, 16]. Besides, we found a slightly higher prevalence of adverse events among previously infected participants similar to a large-scale study [10]. It is hypothesized that this increased reactogenicity may be due to higher antibody titers and increased immunogenicity in previously infected individuals [20].

Among our participants, almost 10% reported a post-vaccination SARS-CoV-2 infection and most of them occurred after taking the first dose of the vaccine. However, no vaccine breakthrough COVID-19 was reported. A recent study among Indian health care workers reported that almost 36% of the total SARS-CoV-2 infection was post-vaccination infection and most of these occur after the first dose [18]. The Center for Disease Control of the USA (CDC), reported 3,729 vaccine breakthrough cases out of 144 million vaccinated people by June 2021 [21]. The emergence of newer variants of the vaccine might be responsible for these vaccine breakthrough cases which warrant further investigation [4]. Long-term follow-up is necessary to snap the actual picture of vaccine breakthrough infection among the population.

The main strength of this study lies in the fact that it provides a comparative picture of the side effects that occurred after taking the ChAdOx1 nCoV-19 and BBIBP-CorV vaccine among the HCPs of Nepal. Despite this fact, there are some limitations of the study. First of all, the study included only a small number of HCPs based on convenience sampling. This could potentially exclude a good number of individuals with severe side effects. Besides, we could not verify the membership of the participants as the HCWs. Moreover, the small sample size of the present study might not be representative of the large population of HCPs in Nepal. Moreover, information about the geographic location and ethnic characteristics of the participants were not collected. Hence, the prevalence of vaccine-related adverse events found in the present study might not be generalizable for the overall population of the country. Being a self-reported study, recall bias could not be rolled out. Besides, the post-vaccination infection may not be generalized to the overall population as we know that healthcare workers are tested more frequently even if they are asymptomatic. Finally, we could investigate only the short-term side effects of the vaccine. A sustainable and long-term surveillance method in the general population is necessary to evaluate possible future adverse events and remedies.

## Conclusions

Adverse events of both vaccines deployed in Nepal are mostly constitutional symptoms like injection site pain and tenderness, headache, fever, and fatigue, and are mild in severity. The ChAdOx1 nCoV-19 vaccine recipients reported a slightly higher rate of adverse events

compared to the BBIBP-CorV vaccine recipients. The finding should be widely disseminated among mass people to make them more confident in vaccine uptake and reduce vaccine hesitancy in general.

## Supporting information

**S1 Dataset.**
(XLSX)

## Acknowledgments

The authors would like to thank all the participants for their cooperation in conducting the study.

## Author Contributions

**Conceptualization:** Sushil Rayamajhi, Md. Abdur Rafi, Pranay Shrestha.

**Data curation:** Md. Abdur Rafi, Anjana Singh Dongol.

**Formal analysis:** Md. Abdur Rafi, Anjana Singh Dongol, Subhechchha Bhandari, Md. Golam Hossain.

**Investigation:** Shreejana Rayamajhi.

**Methodology:** Sushil Rayamajhi, Pranay Shrestha, M. Tasdik Hasan, Md. Golam Hossain.

**Resources:** Nishant Tripathi, Shreejana Rayamajhi, Subhechchha Bhandari, M. Tasdik Hasan, Md. Golam Hossain.

**Software:** Md. Abdur Rafi.

**Supervision:** Minalma Pandey, Md. Golam Hossain.

**Validation:** Nishant Tripathi, Anjana Singh Dongol, Shreejana Rayamajhi, M. Tasdik Hasan, Md. Golam Hossain.

**Visualization:** Nishant Tripathi, Minalma Pandey, M. Tasdik Hasan, Md. Golam Hossain.

**Writing – original draft:** Sushil Rayamajhi, Md. Abdur Rafi, Shreejana Rayamajhi, M. Tasdik Hasan, Md. Golam Hossain.

**Writing – review & editing:** Sushil Rayamajhi, Md. Abdur Rafi, Nishant Tripathi, Anjana Singh Dongol, Minalma Pandey, Shreejana Rayamajhi, Subhechchha Bhandari, Pranay Shrestha, M. Tasdik Hasan, Md. Golam Hossain.

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
