## [Decision Letter · Decision Letter 0]

14 Apr 2022

PONE-D-21-28369Adverse events following immunization with ChAdOx1 nCoV-19 and BBIBP-CorV vaccine: a real-world, comparative study among healthcare professionals of NepalPLOS ONE

Dear Author,

Thank you for submitting your manuscript to PLOS ONE. After careful consideration, we feel that it has merit but does not fully meet PLOS ONE’s publication criteria as it currently stands. Therefore, we invite you to submit a revised version of the manuscript that addresses the points raised during the review process. The reviewer comments are available below for your reference. 

We look forward to receiving your revised manuscript.

Kind regards,

Subish Palaian

Academic Editor

PLOS ONE

3. Please amend your current ethics statement to address the following concerns:

a) Did participants provide their written or verbal informed consent to participate in this study?

Additional Editor Comments:

Dear author,

Regards.

The manuscript has been reviewed by two reviewers and the comments are available below. The comments are mainly related to the methodology and data analysis. Kindly revise the manuscript based on the comments.

Best wishes

Subish Palaian

Reviewers' comments:

Reviewer's Responses to Questions

**Comments to the Author**

1. Is the manuscript technically sound, and do the data support the conclusions?

Reviewer #1: Partly

Reviewer #2: Partly

2. Has the statistical analysis been performed appropriately and rigorously? 

Reviewer #1: I Don't Know

Reviewer #2: Yes

3. Have the authors made all data underlying the findings in their manuscript fully available?

Reviewer #1: No

Reviewer #2: Yes

4. Is the manuscript presented in an intelligible fashion and written in standard English?

Reviewer #1: Yes

Reviewer #2: No

5. Review Comments to the Author

Reviewer #1: Thank you for the invitation to review this manuscript. The manuscript is mostly well-written and presented. The study is interesting and relevant. A few corrections to language may be required at certain places.

My concerns are related to the Methodology of the study. The population of health care professionals in Nepal is large and a sample of 606 health care professionals (HCPs) may not adequately represent this population. The sampling method in this study was a convenience one as stated by the authors. I am not sure that this sample can adequately provide data from the population of HCPs in Nepal. The authors have provided no information about the geographical distribution of the respondents within the country. Does the sample correspond to the population of HCPs in Nepal regarding demographic characteristics? I am not sure that the conclusions as mentioned in the abstract follow from the study results. The conclusions in the main manuscript seem OK. As the denominator is small I am not sure prevalence of adverse effects can be calculated. The findings will be only the prevalence among the study sample, and it will be difficult to generalize it to the general population of HCPs in Nepal.

In the Results can the authors explain why they mention no vaccine breakthrough infection was noted. The authors have not mentioned about IRB approval and obtaining written, informed consent from the participants in this study. I would recommend specialised statistical review of this manuscript.

The authors should deposit the de-identified data in a publicly available repository.

Reviewer #2: The findings from this study will be of importance to advocate the public on vaccination and reduce their anxiety regarding the side effects.

Overall, please check the grammatical errors and paraphrasing needs to be done as per recommendation.

Results

1) The first Chi-square analysis seems unnecessary as the parameters are not associated with the dosing completion since the vaccination is dependent upon availability and the set duration between first and second dose.

2) The graphs need legends and appropriate choice of graph type is imperative.

Conclusion

This article has been clearly written however the conclusion about the anaphylaxis being rare is a bit of a stretch as this has not been evaluated in this study.

6. PLOS authors have the option to publish the peer review history of their article (what does this mean?). If published, this will include your full peer review and any attached files.

Reviewer #1: No

Reviewer #2: No

---

## [Author Response · Author response to Decision Letter 0]

12 May 2022

Response to Reviewers Date: April 30, 2022

Paper Title: Adverse events following immunization with ChAdOx1 nCoV-19 and BBIBP-CorV vaccine: a real-world, comparative study among healthcare professionals of Nepal 

Journal Name: PLOS ONE

Paper ID: PONE-D-21-28369

Dear Editor

Thank you very much for providing reviewers’ insightful remarks on our manuscript. We have made the necessary changes and revised the manuscript accordingly, and detailed point–by–point corrections are given below:

Review Reports:

Reviewer #1: 

Thank you for the invitation to review this manuscript. The manuscript is mostly well-written and presented. The study is interesting and relevant. A few corrections to language may be required at certain places.

Authors reply: We would like to thank the reviewer for the kind words. We have revised the grammatical errors according to your suggestion.

My concerns are related to the Methodology of the study. The population of health care professionals in Nepal is large and a sample of 606 health care professionals (HCPs) may not adequately represent this population. The sampling method in this study was a convenience one as stated by the authors. I am not sure that this sample can adequately provide data from the population of HCPs in Nepal. The authors have provided no information about the geographical distribution of the respondents within the country. Does the sample correspond to the population of HCPs in Nepal regarding demographic characteristics? I am not sure that the conclusions as mentioned in the abstract follow from the study results. The conclusions in the main manuscript seem OK. As the denominator is small I am not sure prevalence of adverse effects can be calculated. The findings will be only the prevalence among the study sample, and it will be difficult to generalize it to the general population of HCPs in Nepal.

Authors reply: We have calculated the sample size for the present study from a previously published article. However, we agree with you that, the sample size included in the present study might not be representative of the large population of the HCPs in Nepal. Moreover, geographic and ethnic variety was also not considered in the present study which limits the generalizability of the findings. We have mentioned these issues in the limitation section of the manuscript (Page 13, Line 223-227).

“Moreover, the small sample size of the present study might not be representative of the large population of HCPs in Nepal. Moreover, information about the geographic location and ethnic characteristics of the participants were not collected. Hence, the prevalence of vaccine-related adverse events found in the present study might not be generalizable for the overall population of the country.”

In the Results can the authors explain why they mention no vaccine breakthrough infection was noted. 

Authors reply: Thanks for mentioning the issue. We have added the definition of vaccine breakthrough infection as an explanation of the statement which was missing in the previous version (Page 10, Line 174-175). 

“No vaccine breakthrough infection which is defined as infection by SARS-CoV-2 after at least 14 days of completion of two-dose vaccination was reported in the study.”

The authors have not mentioned about IRB approval and obtaining written, informed consent from the participants in this study. I would recommend specialised statistical review of this manuscript.

Authors reply: Thanks for mentioning the issue. The statement regarding ethical considerations is now mentioned in the manuscript (Page 6. Line 122-125).

“The ethical clearance of the research proposal was obtained from the Nepal Health Research Council (NHRC) (Reference no. 351). Informed written consent was obtained through email from the participants who agreed to participate in the study as explained on the first page of the survey form.”

The authors should deposit the de-identified data in a publicly available repository.

Authors reply: Thanks for your suggestion. The data will be made publicly available. 

Reviewer #2: 

The findings from this study will be of importance to advocate the public on vaccination and reduce their anxiety regarding the side effects.

Overall, please check the grammatical errors and paraphrasing needs to be done as per recommendation.

Authors reply: We would like to thank the reviewer for the kind words. We have revised the grammatical errors according to your suggestion.

Results

1) The first Chi-square analysis seems unnecessary as the parameters are not associated with the dosing completion since the vaccination is dependent upon availability and the set duration between first and second dose.

Authors reply: Thanks for your suggestion. We have omitted the Chi-square analysis from Table 1.

2) The graphs need legends and appropriate choice of graph type is imperative.

Authors reply: Thanks for your suggestion. We have added legends in the graphs.

Conclusion

This article has been clearly written however the conclusion about the anaphylaxis being rare is a bit of a stretch as this has not been evaluated in this study.

Authors reply: Thanks for your valuable comment. We agree with you and deleted the statement from the conclusion (Page 13, Line 235). 

We would like to thank the reviewers for the valuable comments. We have revised the documents to the best of our ability, but we will definitely be happy to provide further improvement if there are further clarifications required. 

With best regards

Dr. Md. Golam Hossain

Professor of Health Research Group

Department of Statistics, University of Rajshahi

Rajshahi-6205, Bangladesh

E-mail: hossain95@yahoo.com

---

## [Decision Letter · Decision Letter 1]

20 Jul 2022

PONE-D-21-28369R1Adverse events following immunization with ChAdOx1 nCoV-19 and BBIBP-CorV vaccine: a real-world, comparative study among healthcare professionals of NepalPLOS ONE

Dear Dr. Hossain,

Thank you for submitting your manuscript to PLOS ONE. After careful consideration, we feel that it has merit but does not fully meet PLOS ONE’s publication criteria as it currently stands. Therefore, we invite you to submit a revised version of the manuscript that addresses the points raised during the review process.

ACADEMIC EDITOR:The second reviewer from your first round of review has not responded, so we had to use a third reviewer.

We have noted that the reviewer has given citation suggestions, whether or not you follow the suggestions will not affect the final decision. Consider only those that fit your research topic.

We look forward to receiving your revised manuscript.

Kind regards,

Kovy Arteaga-Livias

Academic Editor

PLOS ONE

Journal Requirements:

Reviewers' comments:

Reviewer's Responses to Questions

**Comments to the Author**

1. If the authors have adequately addressed your comments raised in a previous round of review and you feel that this manuscript is now acceptable for publication, you may indicate that here to bypass the “Comments to the Author” section, enter your conflict of interest statement in the “Confidential to Editor” section, and submit your "Accept" recommendation.

Reviewer #1: All comments have been addressed

Reviewer #3: All comments have been addressed

2. Is the manuscript technically sound, and do the data support the conclusions?

Reviewer #1: Yes

Reviewer #3: Yes

3. Has the statistical analysis been performed appropriately and rigorously? 

Reviewer #1: Yes

Reviewer #3: Yes

4. Have the authors made all data underlying the findings in their manuscript fully available?

Reviewer #1: Yes

Reviewer #3: Yes

5. Is the manuscript presented in an intelligible fashion and written in standard English?

Reviewer #1: Yes

Reviewer #3: Yes

6. Review Comments to the Author

Reviewer #1: Kindly deposit the data associated with the study in a public repository. Otherwise the authors have addressed the comments.

Reviewer #3: Title:

I suggest removing the title of "Real world". The instrument in this case was designed specifically for this study.

Summary:

Avoiding the term "fully", currently there are people with 3 or even 4 doses of vaccination.

Introduction.

I suggest considering an additional base bibliography. https://doi.org/10.54034/mic.e1262

Methods.

There was some way to confirm that the respondents were health personnel. The instrument was online, so how could membership in the group of HCWs be verified?

Results.

Avoid the use of "partially" and "Fully". Indicate 1 or two doses.

Were there subjects with a heterologous vaccine schedule?

Discussion.

Add a paragraph about the possible presence of variants during the study period. https://doi.org/10.54034/mic.e1256

7. PLOS authors have the option to publish the peer review history of their article (what does this mean?). If published, this will include your full peer review and any attached files.

Reviewer #1: **Yes: **Pathiyil Ravi Shankar

Reviewer #3: No

---

## [Author Response · Author response to Decision Letter 1]

23 Jul 2022

Response to Reviewers Date: July 23, 2022

Paper Title (old): “Adverse events following immunization with ChAdOx1 nCoV-19 and BBIBP-CorV vaccine: a real-world, comparative study among healthcare professionals of Nepal” 

Paper title (New): “Adverse events following immunization with ChAdOx1 nCoV-19 and BBIBP-CorV vaccine: a comparative study among healthcare professionals of Nepal”

Journal Name: PLOS ONE

Paper ID: PONE-D-21-28369R1

Dear Editor

Thank you very much for providing reviewers’ insightful remarks on our manuscript. We have made the necessary changes and revised the manuscript accordingly, and detailed point–by–point corrections are given below:

Review Reports:

Reviewer #1: 

Kindly deposit the data associated with the study in a public repository. Otherwise the authors have addressed the comments.

Authors’ reply: Thank you for your comment. We have already submitted the dataset to the Journal repository during manuscript submission.

Reviewer #3: 

Title:

I suggest removing the title of "Real world". The instrument in this case was designed specifically for this study.

Authors’ reply: Thank you for your suggestion. According to your suggestion, we have removed “Real world” from the title. The new title is, “Adverse events following immunization with ChAdOx1 nCoV-19 and BBIBP-CorV vaccine: a comparative study among healthcare professionals of Nepal”

Summary:

Avoiding the term "fully", currently there are people with 3 or even 4 doses of vaccination.

Authors’ reply: We have avoided the term “fully” [Line: 46]. 

Introduction:

I suggest considering an additional base bibliography. https://doi.org/10.54034/mic.e1262

Authors’ reply: We have added the suggested bibliography [reference 4].

Methods:

There was some way to confirm that the respondents were health personnel. The instrument was online, so how could membership in the group of HCWs be verified?

Authors’ reply: Thank you for your comment. We have circulated the online questionnaire within the professional social media groups of the HCWs. However, we did not take any specific measure to verify their identity as HCWs which we have mentioned in the limitation section [Page 13, line 234-235].

Results.

Avoid the use of "partially" and "Fully". Indicate 1 or two doses.

Were there subjects with a heterologous vaccine schedule?

Authors’ reply: Thank you for the suggestion. We have corrected the manuscript accordingly [Table 1, Page, 7; Line:138-141].

Discussion.

Add a paragraph about the possible presence of variants during the study period. https://doi.org/10.54034/mic.e1256

Authors’ reply: Thank you for the suggestion. We have added the statement in the discussion [Page 12, line 215-216]

We would like to thank the reviewers for the valuable comments. We have revised the documents to the best of our ability, but we will definitely be happy to provide further improvement if there are further clarifications required. 

With best regards

Dr. Md. Golam Hossain

Professor of Health Research Group

Department of Statistics, University of Rajshahi

Rajshahi-6205, Bangladesh

E-mail: hossain95@yahoo.com

---

## [Editor Report · Decision Letter 2]

26 Jul 2022

Adverse events following immunization with ChAdOx1 nCoV-19 and BBIBP-CorV vaccine: a comparative study among healthcare professionals of Nepal

PONE-D-21-28369R2

Dear Dr. Hossain,

We’re pleased to inform you that your manuscript has been judged scientifically suitable for publication and will be formally accepted for publication once it meets all outstanding technical requirements.

Kind regards,

Kovy Arteaga-Livias

Academic Editor

PLOS ONE
---

## [Editor Report · Acceptance letter]

2 Aug 2022

PONE-D-21-28369R2 

Adverse events following immunization with ChAdOx1 nCoV-19 and BBIBP-CorV vaccine: a comparative study among healthcare professionals of Nepal 

Dear Dr. Hossain:

I'm pleased to inform you that your manuscript has been deemed suitable for publication in PLOS ONE. Congratulations! Your manuscript is now with our production department. 

Kind regards, 

on behalf of

Dr. Kovy Arteaga-Livias 

Academic Editor

PLOS ONE